# Coordinated Diel Gene Expression of Cyanobacteria and Their Microbiome

**DOI:** 10.3390/microorganisms9081670

**Published:** 2021-08-05

**Authors:** Kai Wang, Xiaozhen Mou

**Affiliations:** Department of Biological Sciences, Kent State University, Kent, OH 44242, USA; kwang12@kent.edu

**Keywords:** *Microcystis*, cyanobacterial blooms, diel transcription, microbiome

## Abstract

Diel rhythms have been well recognized in cyanobacterial metabolisms. However, whether this programmed activity of cyanobacteria could elicit coordinated diel gene expressions in microorganisms (microbiome) that co-occur with cyanobacteria and how such responses in turn impact cyanobacterial metabolism are unknown. To address these questions, a microcosm experiment was set up using Lake Erie water to compare the metatranscriptomic variations of *Microcystis* cells alone, the microbiome alone, and these two together (whole water) over two day-night cycles. A total of 1205 *Microcystis* genes and 4779 microbiome genes exhibited significant diel expression patterns in the whole-water microcosm. However, when *Microcystis* and the microbiome were separated, only 515 *Microcystis* genes showed diel expression patterns. A significant structural change was not observed for the microbiome communities between the whole-water and microbiome microcosms. Correlation analyses further showed that diel expressions of carbon, nitrogen, phosphorous, and micronutrient (iron and vitamin B_12_) metabolizing genes were significantly coordinated between *Microcystis* and the microbiome in the whole-water microcosm. Our results suggest that diel fluxes of organic carbon and vitamin B_12_ (cobalamin) in *Microcystis* could cause the diel expression of microbiome genes. Meanwhile, the microbiome communities may support the growth of *Microcystis* by supplying them with recycled nutrients, but compete with *Microcystis* for iron.

## 1. Introduction

Cyanobacteria are a group of oxygenic photosynthetic prokaryotes; they possess many adaptive strategies, which give them competitive advantages over other primary producers in aquatic environments [1]. One of such critical adaptations is their temporal partitioning of cellular metabolisms according to daily fluctuations of light [2], i.e., performing light-dependent and energy-consuming biosynthesis (anabolism) mainly during the daytime and generating energy by breaking down synthesized organic molecules (catabolism) mainly at night [3]. The discovery and knowledge of cyanobacterial diel activities have been mainly based on studies of marine species, and have identified *kaiABC* as the core regulatory gene [4]. One recent study in Lake Erie reported diel metabolic functions in freshwater cyanobacteria *Microcystis* [5]. However, in general, studies on diel activities of freshwater cyanobacterial species are still rare.

Bacterioplankton is a universal and long-term partner of cyanobacteria in aquatic environments [6,7]. Some co-occurring bacteria are attached to cyanobacterial cells [8,9], whereas others are free-living [10]. The activities of these cyanobacterial microbiome species can impact the function and structures of cyanobacterial communities [11]. Microbiome organisms may regenerate essential inorganic nutrients to promote the growth of cyanobacteria [12]. Meanwhile, microbiome organisms can obtain labile organic substrates from cyanobacteria to fulfill their heterotrophic living [10]. While the diel expression of genes is well recognized for cyanobacteria [2,5], the role of the cyanobacterial microbiome in this process remains largely unknown.

Diel gene expression has rarely been studied in bacterioplankton, even though this phenomenon is believed to be universal for all three domains of life [13]. Relevant studies have become available only recently, and are mostly restricted to marine environments [14,15,16,17]. These studies have found that marine bacterioplankton exhibit a diel expression of the genes that are involved in a variety of functions, most of which are involved in nitrogen and phosphorus recycling [14,15,16,17], iron utilization [17], and vitamin B_12_ biosynthesis [15]. However, the driving forces behind the diel expressions of bacterial genes and their impact on cyanobacterial activities are unclear.

This study aimed to further our understanding of the mechanisms governing the diel gene expressions of cyanobacteria and their microbiome. We hypothesize that cyanobacteria can elicit coordinated diel gene expressions in their microbiome organisms, which may in turn impact the diel metabolism of cyanobacteria. Microcosm experiments were set up using surface water collected from Lake Erie, a Laurentian Great Lake that suffers from annual CyanoHABs [18]. The diel dynamics of the metatranscriptomes of whole water (*Microcystis* and microbiome), *Microcystis* enrichment, and microbiome enrichment samples were examined every 12 h over a two-day period (Appendix A). Comparisons of the gene expression patterns between different types of microcosms allowed for the identification of diurnally expressed genes of cyanobacteria and their microbiomes.

## 2. Materials and Methods

### 2.1. Microcosm Experiment Set Up

Surface water samples (~0.5 m) were collected around noon from the western basin of Lake Erie (41.686°, −83.378°) using three sterilized 20 L sample containers during an ongoing *Microcystis* bloom on 26 July 2019. The water samples were immediately placed on ice and were transported in the dark back to the lab within four hours. Once they arrived, the water samples were transferred to a sterilized (washed by 70% ethanol) glass aquarium and incubated at room temperature under natural light for 2 h without agitation, which resulted in a layer of concentrated *Microcystis* colonies (i.e., scum) at the top surface of the water (confirmed by microscopic analysis). Surface *Microcystis* scum (~5 cm) was collected and then filtered sequentially through 25.0 μm nylon net filters and 0.2 μm pore size membrane filters to separate the *Microcystis* colonies and their microbiome communities. The filtrates that passed through 0.2 μm pore size membrane filters were stored in acid-washed bottles and were referred to as the filter-sterilized lake water for subsequent incubation experiments. *Microcystis* colonies collected on the 25.0 μm nylon net filters (*n* = 5) were further washed with 100 mL of the filter-sterilized lake water five times to remove free-living and loosely attached microbes; the washed *Microcystis* cells were transferred to a glass aquarium (length of 50 cm, width of 25 cm, and height of 30 cm) filled with 10 L filter-sterilized lake water and designated as the *Microcystis* enrichment (MCY). The microbiome organisms collected on the 0.2 μm pore size membrane filters combined with the washed microbes were transferred to another glass aquarium, mixed with 10 L filter-sterilized lake water, and designated as the microbiome enrichment (MIB). The MIB was incubated in constant darkness conditions for three days before subsequent incubation to reduce the viability of the phototrophs. The whole water (WW) control was set up in a third glass aquarium using 10 L unprocessed raw water. All three aquaria were cleaned with 5% hydrochloric acid and Milli-Q water before use.

The microcosms were pre-incubated for 72 h before collecting the first set of samples to allow the microbial communities to acclimate. During pre-incubation, the aquaria for the whole water control and *Microcystis* enrichment were incubated in the Kent State University greenhouse uncovered under the natural light-dark cycle at ~27 °C. The microbiome enrichment was incubated under the same temperature, but in constant darkness conditions to reduce the viability of the phototrophs [19]. From the sample collection point, all three aquaria were incubated under the same condition (natural light-dark cycle at ~27 °C). No obvious water loss was observed during the entire experiment. The water in each aquarium was stirred gently every 6 h with a sterilized glass stir stick. After pre-incubation, two water samples (250 mL each) were collected every 12 h from each aquarium for a total of two days. Before sample collection, the water in each aquarium was mixed gently using a sterilized glass stir stick and then measured for water temperature (Temp), pH, dissolved oxygen (DO), and electronic conductivity (EC) using a multiparameter meter (YSI, ProODO 626281, Yellow Springs, OH, USA) and a portable pH/conductivity meter (Thermo Orion, StarA325, Frederick, MD, USA).

After collection, the water samples were immediately filtered through 0.22 μm pore size membrane filters. The filters were stored in a 1.5-mL microtube that contained a 600 μL of RLT plus solution with the RNase inhibitor (QIAGEN, Chatsworth, CA, USA), and were stored at −80 °C immediately before RNA extraction. The filtrates (passed 0.22 μm membrane) were collected in 50 mL sterile conical centrifuge tubes and stored at −20 °C until analyses of the nitrate (NO_3_^−^), nitrite (NO_2_^−^), ammonium (NH_4_^+^), and soluble reactive phosphorus (SRP).

### 2.2. Nutrient Analyses

The nutrients were measured following the procedures described in the standard EPA methods. Briefly, concentrations of NO_3_^−^, NO_2_^−^, NH_4_^+^, and SRP were determined based on the automated hydrazine reduction method (Ohio EPA Method 4500), phenate method (Ohio EPA method 4500B), and colorimetry method (USEPA method 365.1), respectively. The NO_3_^−^, NO_2_^−^, and NH_4_^+^ concentrations were measured using membrane-suppression ion chromatography (Dionex, Thermo Scientific, Carlsbad, CA, USA), and the SRP concentrations were quantified using spectrophotometer DU730 (Beckman coulter, Brea, CA, USA).

### 2.3. RNA Extraction, Library Preparation, and Sequencing

RNA was extracted following the protocol described in Wang et al. (2021) [20]. Briefly, the filters were disrupted and lysed using a Mini-Beadbeater-16 (BioSpec Products, Inc., Bartlesville, OK, USA) twice for 30 s and then centrifuged at 10,000 g for 3 min. The supernatant was then transferred to a new sterile tube, and the RNA was extracted and purified using the AllPrep DNA/RNA kit (QIAGEN). The genomic DNA was removed using the TURBO DNA-free kit (Life Technologies, Foster City, CA, USA). For each sample, approximately 200 ng of purified RNA was treated with Ribo-Zero (Illumina, San Diego, CA, USA) to remove the ribosomal RNA (rRNA) and was then purified using the RNeasy kit (QIAGEN). The samples were confirmed to be free of bacterial DNA by PCR using primers 27F and 1522R. Complementary DNA (cDNA) synthesis was performed using the SuperScript double-strand cDNA synthesis kit (Life Technologies). The cDNA was purified using an Illumina NexteraXT DNA library prep kit according to the manufacturer’s instruction (Illumina). Sequencing was performed on a HiSeq platform for 150 bp paired-end reads (Illumina) in Genewiz. The raw reads were deposited in the National Center for Biotechnology Information (NCBI) short read archive database under BioProject number SRR13450943-SRR13450964.

### 2.4. Sequence Analysis

Low-quality reads (Phred score <20, reads <100 bp, and adaptor contaminants) were removed using Trimmomatic version 0.39 [21]. The ribosomal RNA (rRNA) reads were removed using SortMeRNA version 2.1 [22], and the remaining mRNA reads were used in the downstream analyses.

To examine the transcriptomic patterns of *Microcystis*, the mRNA reads were first mapped to *Microcystis aeruginosa* NIES 843 genome (accession number: NC_010296.1) using STAR software version 2.7.3 [23]. The remaining unmapped reads from each sample were assembled through de novo assembly using Trinity software [24]. Individual assemblies of all of the samples were then pooled together and clustered at 98% identity [15] using CD-Hit [25] to combine the highly similar contigs across the samples. The merged contigs were then filtered to remove sequences shorter than 200 nucleotides and rRNA reads, and were then translated into the corresponding amino acid sequences using Prodigal software [26].

Taxonomic affiliation of the contigs was assigned based on Kaiju software version 1.7.2 using a comprehensive protein database (including Archaea, Bacteria, Eukaryotes, and Viruses) [27]. A resampling procedure was performed before running the Kaiju software based on the sample that had the lowest number of sequences in order to ensure the results were comparable between samples. Functional annotations of translated amino acid sequences were obtained by DIAMOND BLASTP [28] against the UniRef90 database [29]. Translated amino acid sequences were also annotated by the Kyoto Encyclopedia of Genes and Genomes (KEGG) with the online Automatic Annotation Server using the single-directional best-hit method targeted to prokaryotes and with the metagenomic option selected.

Read mapping to contigs was carried out with RSEM [30] and the default settings with the exception of using the paired-end option and the bowtie2 [31] option. The orthologous groups (OGs) were generated by performing a reciprocal comparison with DIAMOND BLASTP followed by MCL (Markov cluster algorithm) set to an inflation parameter of 1.4 following a protocol described previously [15,17]. The read counts were summed across the OGs for the microbiome contigs. For each OG, the taxonomic affiliation of the most abundant contig was used as the taxonomic affiliation for that OG.

### 2.5. Statistical Analysis

To keep the downstream analyses conservative, only those *Microcystis* genes or microbiome OGs that had a total number of reads >100 across all time points were used in the downstream analyses. Statistical comparisons of the *Microcystis* genes and microbiome OGs between a treatment and the control samples (i.e., *Microcystis* genes of the MCY vs. the WW samples, and the microbiome OGs of the MIB vs. the WW samples) were performed using the DESeq2 package [32]. False discovery rate (FDR) values were calculated to correct the corresponding *p* values using the Benjamini–Hochberg algorithm [33]. Genes with FDR corrected *p* values < 0.05 and an absolute fold change ≥1.5 were considered to be differentially expressed.

Count data were normalized using the “varianceStabilizingTransformation” command in the DESeq package [32]. Significant periodicity in normalized *Microcystis* genes and microbiome OGs was determined using rhythmicity analysis incorporating non-parametric methods (RAIN) [34] in R. The *p* values were adjusted by calculating the FDR values. *Microcystis* gene transcripts and microbiome OGs with FDR < 0.1 were considered to have a significant periodicity [15]. To examine the co-expression between *Microcystis* and the microbiome, pairwise Pearson’s correlations between the *Microcystis* gene and microbiome OGs were calculated. The corresponding *p* values were corrected for multiple testing by calculating the FDR values [33].

## 3. Results

### 3.1. Variations of Environmental Conditions

After pre-incubation and throughout the experiment, none of the measured environmental variables had significant temporal variations within any given treatment (Appendix A). However, four out of the eight measured environmental variables showed significant differences among the treatments (Appendix A). Specifically, pH (7.11 in average) and EC (130 μS.cm) had the lowest values in the MCYs and the highest values in the WWs (pH: 7.99; EC: 162 μS.cm) (one-way ANOVA, *p* < 0.05). The DO values were at the same level in the MCYs (8.2 mg/L) and WWs (8.4 mg/L), but were significantly higher in the MIBs (10.1 mg/L) (one-way ANOVA, *p* < 0.05). The NH_4_^+^ values were at the same level between the MIBs (28.3 mg/L) and WWs (25.7 mg/L), but were significantly lower in the MCYs (21.8 mg/L) (one-way ANOVA, *p* < 0.05). The other four variables, i.e., temperature (Temp, 26.9 °C), NO_3_^−^ (27.3 mg/L), NO_2_^−^ (4.85 mg/L), and SRP (17.6 mg/L), showed no significant differences among the treatments (one-way ANOVA, *p* > 0.05; Appendix A).

### 3.2. Diel Expression of Microcystis Genes

A total of 22 metatranscriptomic libraries were generated. After quality control, a total of 669.2 million reads (27.6–44.8 million reads/sample) remained, representing 89.3% of the raw reads. After rRNA removal, each library contained, on average, 17.3 million mRNA reads.

The differential gene expression analysis yielded a total of 506 differentially expressed genes (DEGs) of *Microcystis* between the MCY and WW samples (Appendix A). Among them, 390 were over-expressed and 116 were under-expressed in the MCYs relative to the WWs (Appendix A). RAIN analysis further identified a total of 1885 genes that showed significant diel expression patterns (i.e., either over-expressed in the day or at night) in *Microcystis* (*p* < 0.1) in the MCYs (1195) and WWs (1205); among these diel expression genes in *Microcystis*, 298 were also DEGs and were further designated as diel DEGs (dDEGs) (Appendix A). The three clock genes (*kaiABC*) of *Microcystis* showed significant diel expression patterns, but their expression levels were not significantly different between the MCYs and WWs (Appendix A).

*Microcystis* dDEGs contained 17 genes related to photosynthesis (PS) and carbon fixation, including PS II, PS I, cytochrome b_6_f (Cb_6_f) complex, and the Calvin–Benson–Bassham cycle (CBBC) (Figure 1a–d). Most (15 out of 17) of these *Microcystis* dDEGs maintained the same diel patterns in the MCYs as those in the WWs but their expression levels were higher in the MCYs than the WWs (log_2_FC = 1.6–6.7, *p* < 0.05). The remaining two genes (*pgk*, MAE_30020 of CBBC) had opposite diel patterns in the MCYs and WWs; they peaked at night in the MCYs but peaked during the day in the WWs (Figure 1d). In addition, the expression levels of these CBBC genes were significantly higher in the MCYs than in the WWs (log_2_FC = 1.6–2.9, *p* < 0.05; Figure 1d).

Four of *Microcystis* dDEGs were affiliated with the transformations of macronutrients nitrogen (N) and phosphorus (P), including NH_4_^+^ and nitrate transport, as well as phosphate transport and regulation (Figure 2a). All four of these *Microcystis* dDEGs in the MCYs had the same diel pattern as those in the WWs, with the nitrate transport and phosphate regulator genes peaking during the day, and NH_4_^+^ and phosphate transport genes peaking at night (Figure 2a). The expression levels of the nitrate transport and phosphate regulator genes were significantly higher in the MCYs (log_2_FC = 1.5–4.6, *p* < 0.05) than in the WWs (Figure 2a). In contrast, the expression levels of NH_4_^+^ and the phosphate transport genes were significantly lower (log_2_FC = 2.1–2.7, *p* < 0.05) in the MCYs than in the WWs (Figure 2a).

Seven *Microcystis* dDEGs belonged to micronutrient metabolism, including three for iron (Figure 2b) and four for cobalamin (vitamin B_12_) (Figure 2c). For iron metabolism, the heme oxygenase and ferredoxin genes had the same diel pattern in the MCYs and WWs, both of which peaked during the day. The flavodoxin gene had an opposite diel pattern in the MCYs (peaked at night) from the WWs (peaked during the day; Figure 2b). The expression levels of all three iron metabolism genes were significantly higher in the MCY than in the WW (Figure 2b, log2FC = 1.8–3.2, *p* < 0.05). The diel pattern and expression level of the vitamin biosynthesis genes varied between the MCYs and WWs. The precorrin6 gene exhibited the same diel pattern (peaked at night) in the MCYs and WWs, but its expression levels were significantly lower in the MCYs than in the WWs (log2FC = 1.9–2.8, *p* < 0.05) (Figure 2c). The precorrin4 gene only showed a diel pattern in the MCYs (peaked during the day) but not in the WWs; it was expressed significantly higher in the MCYs (log_2_FC = 2.3–2.8, *p* < 0.05). With significantly lower expression levels (log_2_FC = 1.6–3.4, *p* < 0.05), precorrin8 and NADPH dependent FMN reductase genes lost their diel expression patterns in MCY and only showed diel patterns in the WWs (peaked at night; Figure 2c).

Seven *Microcystis* dDEGs were for stress response (i.e., universal stress protein, chaperone, and glutathione peroxidase genes; Figure 2d) and microcystin synthesis (i.e., *mcyADEG*) (Figure 2e). The universal stress protein (peaked during the day) and chaperone (peaked at night) genes only had diel patterns in the WWs, whereas glutathione peroxidase genes exhibited the same diel pattern (peaked at night) in the MCYs as those in the WWs (Figure 2d). All of these stress response dDEGs were expressed at significantly lower levels (log_2_FC = 2.4–4.9, *p* < 0.05) in the MCYs than in the WWs (Figure 2d). Microcystin synthesis genes (peaked during the day) also exhibited the same diel pattern in the MCYs and WWs (Figure 2e), and their expression levels were significantly higher (log_2_FC = 1.5–2.3, *p* < 0.05) in the former samples (Figure 2e).

### 3.3. Diel Expression of Microbiome Genes

To create one sequence assembly to which the reads from all 12 samples could be mapped onto, the reads from all of the samples were pooled together, which yielded a total of 144,259 contigs. Among these contigs, 6949 were differentially expressed between the MIBs and WWs, and they were designated as microbiome DEGs. Among these microbiome DEGs, 2978 were over-expressed and 3971 were under-expressed in the MIBs relative to the WWs. RAIN analysis identified 4779 microbiome genes that had significant diel expression patterns in the WWs, but none in the MIBs (Appendix A). Among the 4779 diel microbiome genes, 1222 were also DEGs and they were further designated as microbiome dDEGs (Appendix A). None of the dDEGs were identified in MIBs.

Four microbiome dDEGs in the WWs were affiliated with organic carbon (OC) degradation (i.e., carbon-active enzymes (CAZymes), respiration, peptidase, and alkaline phosphatase; Figure 3a,b). CAZymes dDEGs were affiliated with *Bacteroidetes*, while respiration dDEGs were assigned to *Planctomycetes*, *Ciliophora*, and *Zoopagomycota*. Peptidase and alkaline phosphatase dDEGs were affiliated with *Chytridiomycota* and *Ascomycota*, respectively. The CAZymes and respiration dDEGs peaked during the day in the WWs, and their expression levels were significantly higher in the WWs than their expressions in the MIBs (Figure 3a). In contrast, the peptidase and alkaline phosphatase dDEGs peaked at night in the WWs, and their expressions had the same pattern as the CAZymes and respiration dDEGs between the WWs and MIBs (Figure 3b).

Nine microbiome dDEGs in the WWs were related to nutrient transformation (N and P) (Figure 4a–c). These nutrient transformation dDEGs are associated with different taxa. Glutamate dehydrogenase (GDH), NH_4_^+^ transport, P-carrier protein, and P transport dDEGs were assigned to *Mucoromycota* (Fungi), *Discosea* (Protozoa), *Ciliophora*, and *Apicomplexa* (Chromista), respectively. The N regulator protein and deaminase dDEGs were affiliated with *Planctomycetes* (Bacteria), while transaminase, urease accessory protein, and P starvation-inducible protein dDEGs were affiliated with *Proteobacteria* (Bacteria). The GDH, NH_4_^+^ transport, P transport, and P-carrier protein dDEGs peaked at night in the WWs, whereas the rest nutrient transformation microbiome dDEGs peaked during the day in the WWs (Figure 4a–c). The expression levels of GDH, transaminase, P transport, and P carrier protein dDEGs were significantly higher (log_2_FC = 3.1–7.9, *p* < 0.05) in the WWs than the corresponding genes in the MIBs, whereas the P-starvation-inducible protein dDEG was expressed significantly lower (log_2_FC = 1.5–1.6, *p* < 0.05; Figure 4a–c) in the WWs than the corresponding gene in the MIBs. The expression levels of NH_4_^+^, N regulation, deaminase, and urease accessory dDEGs were different from the other nutrient transformation dDEGs, and were only significantly differentially expressed during the day or night. The NH_4_^+^ transport dDEG was expressed at a similar level at night, but was significantly lower (log_2_FC = 1.9, *p* < 0.05) during the day in the WWs than the corresponding gene in the MIBs (Figure 4a). N regulation, deaminase, and urease accessory dDEGs were expressed similarly during the day, but were significantly higher (log_2_FC = 1.5–1.7, *p* < 0.05) at night in the WWs than the corresponding genes in the MIBs (Figure 4a,b).

Five microbiome dDEGs in the WWs were affiliated with the micronutrient metabolism, including four for iron metabolism and one for cobalamin-dependent methionine biosynthesis (Figure 4d). No vitamin synthesis dDEGs or genes were identified in the microbiome. Cytochrome biogenesis and flavodoxin microbiome dDEGs were affiliated with *Proteobacteria* and *Candidatus Omnitrophica*, respectively. Iron-regulated membrane protein, non-heme iron enzyme, and methionine synthase dDEGs were affiliated with *Bacteroidetes*. Cytochrome biogenesis, iron-regulated membrane protein, and methionine biosynthesis dDEGs peaked during the day in the WWs, while the flavodoxin and non-heme iron enzyme dDEGs peaked at night in the WWs (Figure 4d). The expression levels of these micronutrient metabolism dDEGs were all significantly higher (log_2_FC = 1.6–6.9, *p* < 0.05) in the WWs than the corresponding genes in the MIBs (Figure 4d).

Another four microbiome dDEGs in the WWs were associated with stress response (i.e., chaperone, catalase, glutathione peroxidase, and superoxide dismutase) (Figure 4e). The chaperone dDEG was assigned to *Evosea*, while the other dDEGs were affiliated with *Ciliophora*. Chaperone and catalase dDEGs peaked at night in the WWs, while glutathione peroxidase and superoxide dismutase dDEGs peaked during the day in the WWs (Figure 4e). Chaperone, glutathione peroxidase, and superoxide dismutase dDEGs were expressed similarly at night, but significantly higher (log_2_FC = 1.5–2.0, *p* < 0.05) during the day in the WWs than the corresponding genes in the MIBs, whereas catalase dDEG was only expressed significantly higher (log_2_FC = 1.9, *p* < 0.05) at night in the WWs than the corresponding gene in the MIBs (Figure 4e).

The taxonomic structure of the overall microbial community showed no significant diel pattern in any of the treatments (Appendix A). *Microcystis* only accounted for 0.3–0.7% of the microbiome reads in the MIB samples, which was significantly lower than those of the WW samples (87.1%; one-way ANOVA; *p* < 0.05). However, the relative abundances of the 10 major (the relative abundance of microbiome sequences > 0.1%) taxa varied between treatments (Appendix A). The relative abundances of most bacterioplankton (i.e., *Actinobacteria*, *Bacteroidetes*, and *Planctomycetes*) and viruses were significantly higher in the MIBs than in the WWs (one-way ANOVA; *p* < 0.05; Appendix A), except for *Proteobacteria,* which exhibited an opposite pattern (one-way ANOVA; *p* < 0.05; Appendix A). In contrast, most eukaryotes were significantly lower in the MIBs than in the WWs (one-way ANOVA; *p* < 0.05; Appendix A).

### 3.4. Coordinated Diel Gene Expression between Microcystis and the Microbiome

In the WWs, the expression of essential genes to microbial growth was significantly coordinated between *Microcystis* and the microbiome (Figure 5). Specifically, most of the *Microcystis* dDEGs related to PS and C-fixation were positively (0.41 < *r* < 0.84, *p* < 0.05) correlated with the microbiome CZAymes and respiration dDEGs (Figure 5), and were negatively (−0.92 < *r* < −0.61, *p* < 0.05) correlated with the ALP and peptidase dDEGs of the microbiome (Figure 5).

For macronutrient transformation, the *Microcystis* dDEGs related to NH_4_^+^ transport exhibited significant positive correlations with GDH and deaminase dDEGs in the microbiome, while the *Microcystis* dDEG involved in *p* transport had a significant positive correlation with ALP dDEG in the microbiome (Figure 5).

In addition, three cobalamin biosynthesis dDEGs in *Microcystis* exhibited significantly positive correlations (0.51 < *r* < 0.72, *p* < 0.05) with cobalamin-dependent methionine synthase dDEG in the microbiome. For iron metabolism, one *Microcystis* dDEG related to heme oxygenase gene had significant positive correlations (0.63 < *r* < 0.75, *p* < 0.05) with iron-requiring dDEGs (i.e., cytochrome biogenesis and iron regulation membrane protein) in the microbiome (Figure 5).

## 4. Discussion

This work provides one of the first empirical studies to support the hypothesis that the diel metabolic activities in *Microcystis* could elicit a coordinated diel expression in its microbiome [35]. Our study found that less than 50% of *Microcystis* genes that showed a diel expression pattern in the WW samples (with microbiome) remained their diel feature in the MCY samples (without a microbiome), indicating that the presence of microbiome communities had significant impacts on the diel feature of *Microcystis* genes. However, the expression levels of most diel expression *Microcystis* genes (i.e., PS- and iron-related, and MC synthesis genes) in the WW samples were lower than (log_2_FC = 1.5–6.7) those in the MCYs. The exceptions were genes related to NH_4_^+^ transport, and vitamin and peroxidase synthesis were expressed significantly higher in the WWs. In contrast, the microbiome genes only showed diel expression patterns when *Microcystis* existed (WW samples). This suggests that the observed diel expression of the microbiome genes in natural communities [35] was likely a microbial response to the diel activities of *Microcystis*. Our study also found diel oscillation of the microbiome genes not only from heterotrophic bacteria, but also from eukaryotes, including algae, ciliates, and fungi (Appendix A). This broad influence of cyanobacterial diel activities on microbiome organisms may underly the diel fluctuation of the microbial community structure and nutrient flux in the aquatic environments [14,36,37].

In the WWs, the expressions of the *Microcystis* C-fixation genes (Figure 2) were positively correlated with the microbiome genes related to OC-degradation (i.e., CAZymes and respiration, Figure 4b,c), suggesting a potential synchrony between *Microcystis* and the microbiome communities on C metabolism (Figure 6) [15]. The interactions linked to the carbon metabolisms between the autotrophs and associated heterotrophs have been reported in several previous studies [15,16,17], indicating that the associated microorganisms are not products of opportunistic/stochastic colonization [15,16,17]. Specifically, microbiome communities might have gained benefits from the acquisition of OC released by autotrophs (i.e., *Microcystis*) [38] and, in return, they may help lower the alleviated O_2_ stress of *Microcystis* through respiration [39,40]. Meanwhile, our results found that N and P transport genes of *Microcystis* exhibited significant correlations (*r* = 0.69–0.82, *p* < 0.05; Figure 5) with the organic N and P remineralization gene expression of microbiome communities in the WWs. This indicates that *Microcystis* could use inorganic N and P derived from microbiome remineralization (Figure 6), consistent with the results of previous microcosm co-culture experiments by direct measurements [41] and gene transcript analyses [42].

Besides carbon and nutrient metabolism, coordinated diel expression was also found for cobalamin-related activities between cyanobacteria and their microbiome (Figure 2c and Figure 4d). Despite the critical role of the cobalamin in the growth of all organisms, the de novo biosynthesis of cobalamin has only been characterized in selected prokaryotic organisms [43,44]. Based on the genome sequence analysis, *Microcystis aeruginosa* are capable of de novo cobalamin biosynthesis. Three out of four diel expression genes of *Microcystis* were related to cobalamin biosynthesis highly expressed (log_2_FC = 1.6–3.4) in the WWs, and had significant positive correlations with the cobalamin-dependent methionine synthase gene of the microbiome, indicating that *Microcystis* might be the source of cobalamin for its microbiome communities (Figure 6). Similar results were reported by previous studies of marine cyanobacteria *Trichodesmium* and its microbiome [15,40]. These findings consistently suggest that vitamins play important roles in the cyanobacteria-microbiome association, and these compounds could exert a selective pressure on the structures of the cyanobacterial microbiome [15,40]. This might further reinforce that the diel expression of the methionine synthase gene in the microbiome was likely caused by the diel fluxes of cobalamin biosynthesis in *Microcystis*.

Iron is another important co-factor for all of the organisms, and it plays an important role in electron transport [45]. Heme oxygenase has been reported to be involved in liberating iron from organic complexes [46], and this gene in *Microcystis* exhibited a higher expression in the MCYs than WWs (Figure 2c). Meanwhile, several iron-requiring genes in both *Microcystis* and the microbiome showed significant positive correlations with the heme oxygenase gene expression (Figure 5). The significant positive correlations between the heme oxygenase gene and other iron-requiring genes suggest that iron liberated by heme oxygenase during the day could be repurposed for other processes or organisms during the daytime [15]. In addition, diel flavodoxin gene expression was detected in both *Microcystis* (Figure 2b) and the microbiome (Figure 4d), and its expression was higher at night in both *Microcystis* (Figure 2b) and its microbiome (Figure 4d) than during the day. Flavodoxin protein does not require iron, and related genes are usually expressed to substitute for other iron-requiring proteins when the iron is limited [47]. Therefore, the high expression of the flavodoxin gene at night indicates an iron limitation at night for both *Microcystis* and the microbiome communities. Furthermore, the iron-related genes in *Microcystis* were expressed at significantly higher levels in the MCYs than in the WWs (Figure 2b), and microbiome iron-related dDEGs showed an opposite expression pattern (Figure 4d). This suggests an iron competition between *Microcystis* and its microbiome communities in the WWs, which helps to explain the lower expression of PS-related genes (iron serves as a co-factor) in the WWs than MCYs during the day (Figure 2b–d) [17]. Overall, the diel expression pattern of iron-related genes indicates a high demand for iron from both *Microcystis* and microbiome organisms during the daytime, which likely causes competition for this source. A similar conclusion was obtained by a previous marine study, which suggested that *Trichodesmium* and its microbiome share the iron supply [15]. This might impact the photosynthesis of *Microcystis,* as iron is an important photosynthetic co-factor [17].

The expressions of the microcystin (MC) synthesis genes were also detected to follow a significant diel fluctuation in the present study, and the expression levels were lower in the WWs than in the MCYs (Figure 2e). This indicates that the synthesis of MCs was more active without microbiome communities. This disagrees with several previous studies that proposed that cyanotoxins can function as deterrents against grazing [48]. The contrasting findings once again emphasize the complexity of the physiological and ecological role of MCs. MCs have been found to protect C-fixation-related enzymes from oxidative stress [49,50]. This was in line with our finding that MC synthesis genes exhibited the same diel pattern (peaked during the day) as PS and C-fixation-related genes.

The focus of this research was on the impacts of the diel expression of *Microcystis* genes on their microbiome and its reciprocal effect. Therefore, an analysis of the interactions between the cyanobacteria and microbiome were focused on the diel differentially expressed genes (dDEGs) between the samples, although the expression of some non-diel and non-DEG genes might also be coordinated between *Microcystis* and the microbiome communities [51,52,53]. In addition, our microcosm experiments were performed in aquaria that were limited to the size and were subjected to bottle effects; the incubation was performed in a greenhouse, which cannot fully represent the in situ conditions of Lake Erie [54]. This can partially explain the different sets of diel expressed *Microcystis* transcripts found in our study and a previous study in Lake Erie that directly sequenced the transcripts of the total microbial community [5]. Both studies found *Microcystis* transcripts of photosystem II and toxin biosynthesis were enriched in the daytime, however, our study also found that photosystem I and carbon fixation were enriched during the day, opposite from another study [5]. In addition, our study found transcripts of glycogen degradation, vitamin biosynthesis, and detoxifying reactive oxygen species were nighttime enriched; they were not identified as nighttime expressed genes by the other study [5]. Furthermore, our results found that some of the reads in the *Microcystis* enrichment (MCY) samples were assigned to heterotrophic bacteria, which might limit the comparison of the *Microcystis* gene expression between the MCY and WW samples. Moreover, the low replicate numbers for each sample in our study might reduce the statistical power or the confidence of reproducibility. Lastly, the present study is mainly based on gene transcript data, future efforts using metabolomics/proteomics are necessary to further verify and explore the coordinated interactions between the autotrophs and the co-occurring microorganisms. Nevertheless, our approach allowed us to provide one of the first empirical datasets to examine the tightly coupled expression between autotrophs (i.e., *Microcystis*) and their microbiomes.

In conclusion, our study found that the diel metabolic activities in the microbiome communities may be elicited by the diel fluxes in *Microcystis*. The coordinated diel metabolic activities between *Microcystis* and the microbiome communities could be clearly observed in C-processing, nutrient (i.e., N and P) recycling, and vitamin B_12_ supply. Our study also suggests a potential competition of iron between *Microcystis* and its microbiome communities during the daytime, which, in turn, might limit the expression of the PS-related gene expression by *Microcystis*.

## Figures and Tables

**Figure 1 microorganisms-09-01670-f001:**
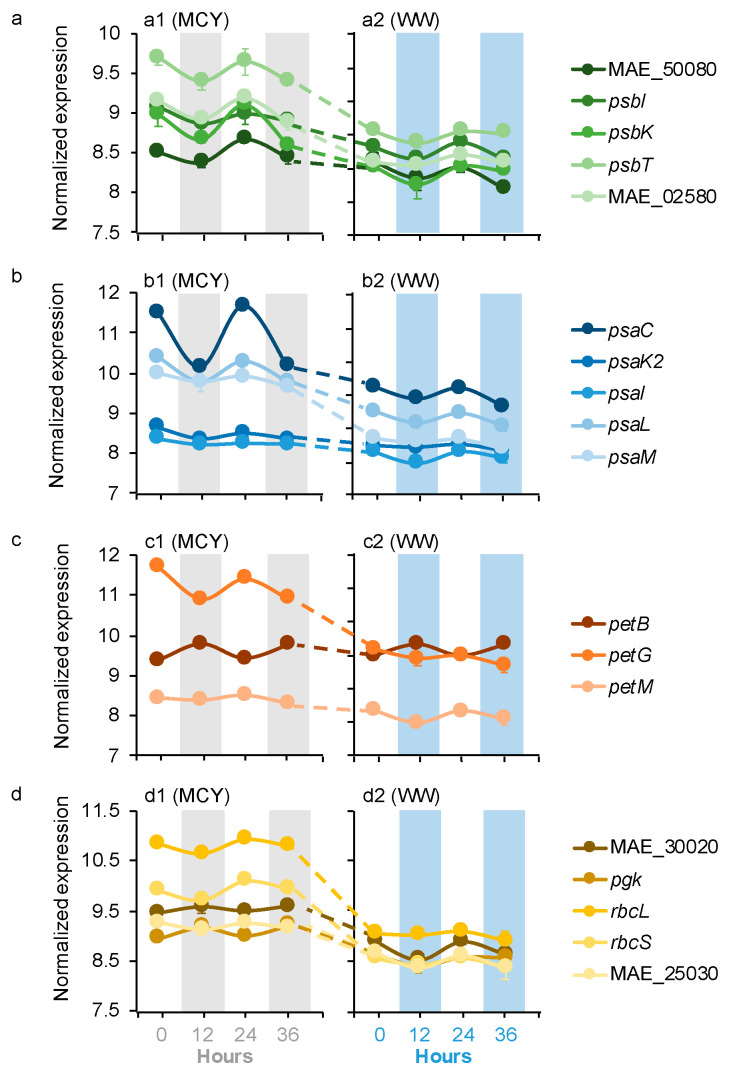
Averages of normalized *Microcystis* gene expression values based on variance stabilizing transformation in DeSeq2 related to (**a**) PSII, (**b**) PSI, (**c**) cb6f, and (**d**) CBBC in the MCY and WW microcosms. Gray and blue bars indicate dark condition samples in the MCY and WW, respectively. The error bar represents the standard error of gene expression values.

**Figure 2 microorganisms-09-01670-f002:**
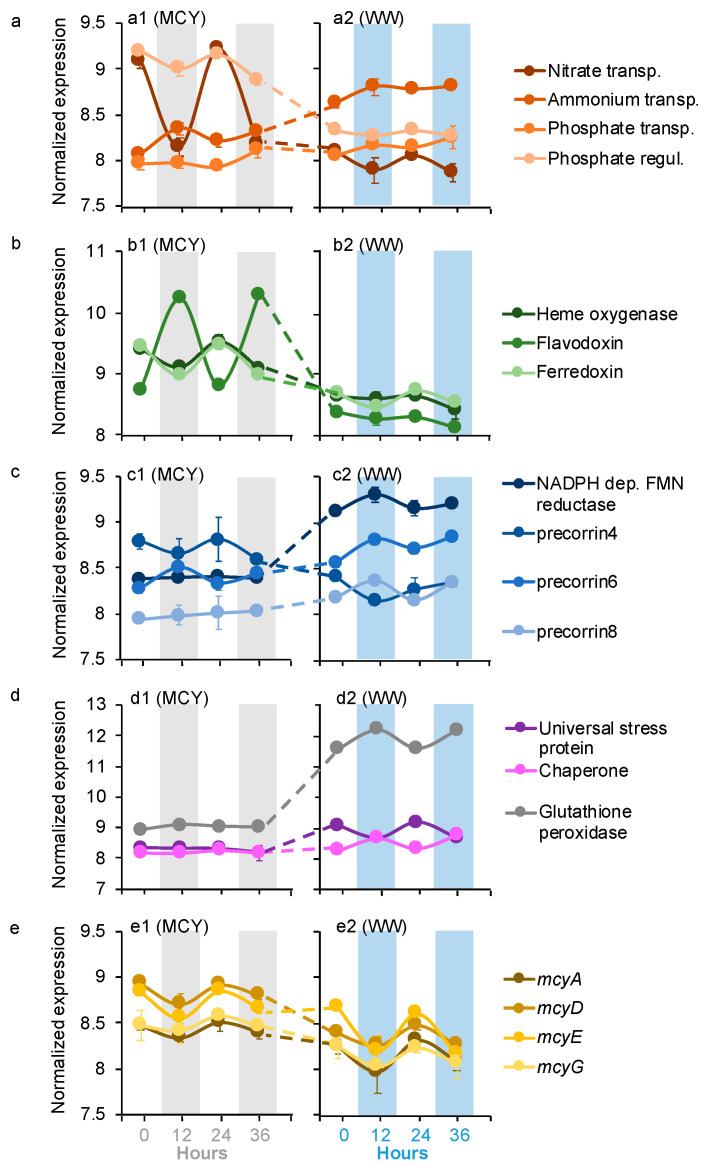
Averages of normalized *Microcystis* gene expression values based on variance stabilizing transformation in DeSeq2 related to (**a**) nitrogen and phosphorus metabolism, (**b**) iron metabolism, (**c**) vitamin biosynthesis, (**d**) stress response and H_2_O_2_ depletion, and (**e**) microcystin synthesis. Gray bars and blue bars indicate dark conditions samples in the MCY and WW, respectively. The error bar represents standard error of the gene expression values.

**Figure 3 microorganisms-09-01670-f003:**
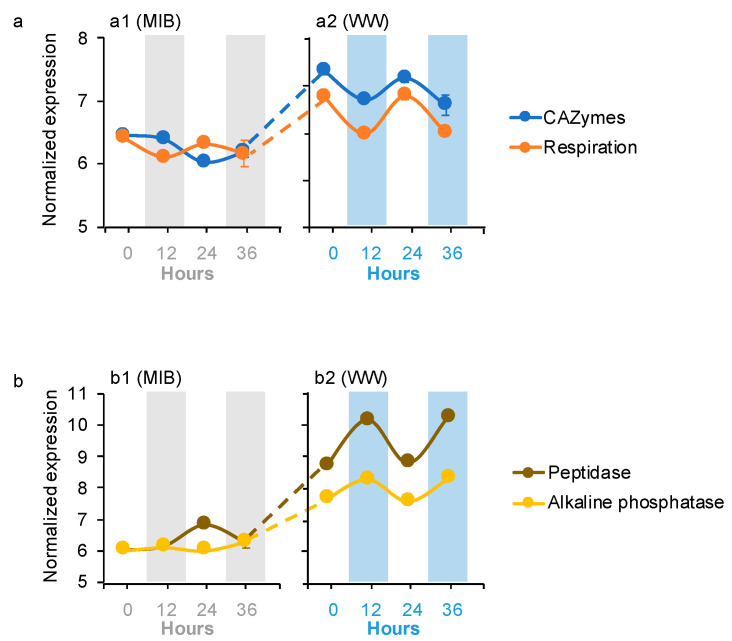
Averages of the normalized microbiome gene expression values based on variance stabilizing transformation in DeSeq2 related to (**a**) carbohydrate-active enzymes and microbiome organism respiration, and (**b**) peptidases and alkaline phosphatase. Gray bars and blue bars indicate dark condition samples in the MIB and WW, respectively. The error bar represents the standard error of the gene expression values.

**Figure 4 microorganisms-09-01670-f004:**
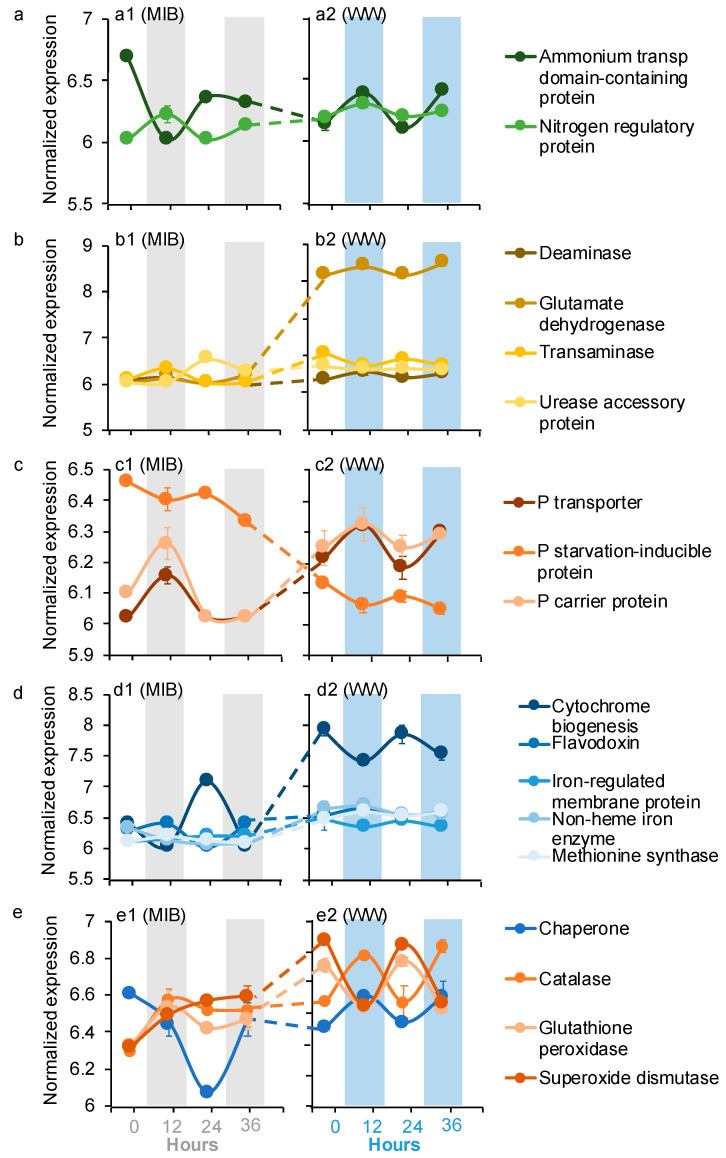
Averages of the normalized microbiome gene expression values based on variance stabilizing transformation in DeSeq2 related to (**a**) nitrogen metabolism, (**b**) organic nitrogen metabolism, (**c**) phosphorus metabolism, (**d**) iron metabolism and methionine synthesis, and (**e**) stress response and H_2_O_2_ depletion. Gray bars and blue bars indicate dark condition samples in MIB and WW, respectively. The error bar represents the standard error of the gene expression values.

**Figure 5 microorganisms-09-01670-f005:**
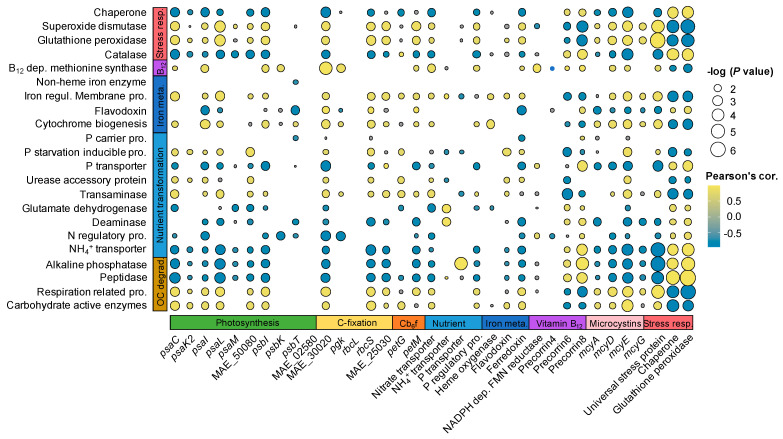
Correlations between *Microcystis* genes (X-axis) and microbiome genes (Y-axis). Only correlations with adjusted *p* < 0.05 are shown.

**Figure 6 microorganisms-09-01670-f006:**
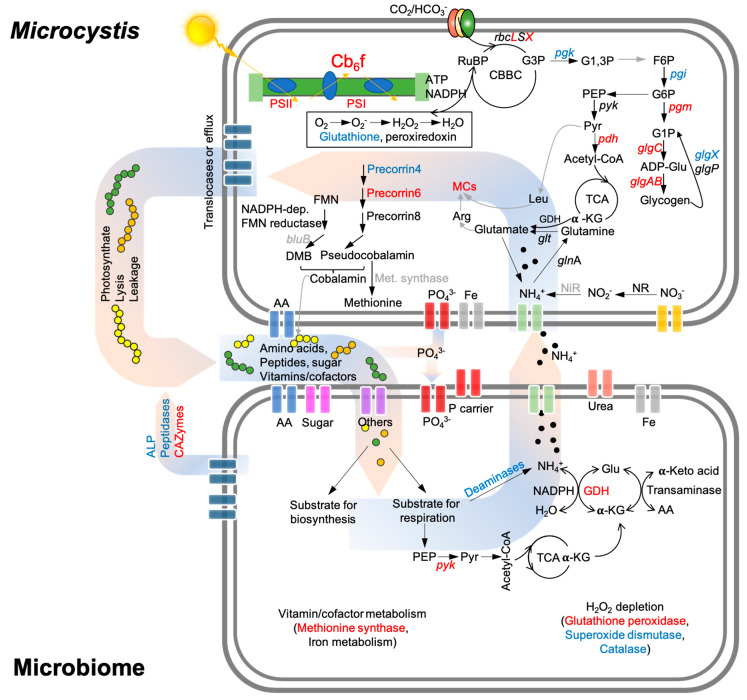
A schematic diagram showing the potential interactions between *Microcystis* and its microbiome communities. AA—amino acids; ALP—alkaline phosphatase; CAZymes—carbohydrate-active enzymes; TCA—tricarboxylic acid cycle; *NR*—nitrate reductase; *NiR*—nitrite reductase; Gln—glutamine; Glu—glutamate; *glnA*—glutamine synthase; *glt* genes—glutamate synthases; GDH—glutamate dehydrogenase; α-KG—alpha-Ketoglutarate; MCs—microcystins; Arg—arginine; Leu—Leucine; *glgABC*—glycogen synthesis; *glgP*—glycogen degradation; *glgX*—Glycogen debranching enzyme; *pgi*—glucose-6-phosphate isomerase; *pgk*—phosphoglycerate kinase; *pgm*—phosphoglucomutase; *pdh*—pyruvate dehydrogenase; *pyk*—pyruvate kinase. The genes that peaked in the daytime and nighttime are in red and blue, respectively.

## Data Availability

The sequence data were submitted to the National Center for Biotechnology Information (NCBI) short read archive database under BioProject number SRR13450943-SRR13450964.

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
