# Peer review of "Coordinated Diel Gene Expression of Cyanobacteria and Their Microbiome"

_microorganisms, 2021, doi:10.3390/microorganisms9081670_

Round 1

Reviewer 1 Report

The paper by Wang and Mou describes diel gene expression in experiments in which a Microcystis bloom was separated into a Microcystis-enriched fraction, a microbiome-enriched fraction and gene expression patterns were compared to a whole water bloom biomass. The authors demonstrate that in the whole water experiment, diel expression was observed in both Microcystis and the microbiome, yet the number of diel differentially expressed genes was considerably lower in the Microcystis-enriched treatment. The work is interesting in that it addresses the importance of cyanobacterial-microbiome interactions in considering overall bloom physiology.

I do have one major concern pertaining to the discussion of the work. There is a paper they mention in one sentence (ref. 5, Davenport) and then subsequently ignore. It seems to me they must present their data in the context of prior work on this very topic (although microbiome gene expression was not part of the Davenport paper). The authors must compare this prior study with theirs an identify areas of agreement and disagreement, and speculate as appropriate why the data sets do not agree. For example, Davenport shows that Photosystem I genes are preferentially expressed at night, as does Kolody et al. 2019 in a marine system. Why is that? The paper presents a nice data set but inadequately discusses its meaning in relation to prior literature.

A minor point - there is no way the sequential filtering and washing of the Microcystis biomass could eliminate all the other microbiome bacteria, so in the Methods please describe the MCY and MIB as enrichments. The paper is written so that it reads like the MCY and MIB preparations are pure cyanobacteria and bacteria-only fractions, and this is certainly not the case.

Author Response

Reviewer #1

The paper by Wang and Mou describes diel gene expression in experiments in which a Microcystis bloom was separated into a Microcystis-enriched fraction, a microbiome-enriched fraction and gene expression patterns were compared to a whole water bloom biomass. The authors demonstrate that in the whole water experiment, diel expression was observed in both Microcystis and the microbiome, yet the number of diel differentially expressed genes was considerably lower in the Microcystis-enriched treatment. The work is interesting in that it addresses the importance of cyanobacterial-microbiome interactions in considering overall bloom physiology.

I do have one major concern pertaining to the discussion of the work. There is a paper they mention in one sentence (ref. 5, Davenport) and then subsequently ignore. It seems to me they must present their data in the context of prior work on this very topic (although microbiome gene expression was not part of the Davenport paper). The authors must compare this prior study with theirs an identify areas of agreement and disagreement, and speculate as appropriate why the data sets do not agree. For example, Davenport shows that Photosystem I genes are preferentially expressed at night, as does Kolody et al. 2019 in a marine system. Why is that? The paper presents a nice data set but inadequately discusses its meaning in relation to prior literature.

Response: The focus of our study and Davenport et al., (2019) were different. Based on the knowledge that cyanobacteria have diel activities, out current study aimed to answer two questions (1) whether the diel activity of cyanobacteria could elicit a coordinated diel gene expression in their co-occurring microorganisms and (2) how responses of the microbiome in turn impact cyanobacterial metabolisms. On the other hand, the paper by Davenport et al. 2019 was focused on diel expression of Microcystis (cyanobacteria) only, but did not study interactions between Microcystis and their co-existing microorganisms.

Therefore, in the present study we focused on examining genes (i.e., nutrients, vitamins, and iron) that have diel expression patterns and played important roles in communications/interactions between cyanobacteria and their microbiome, but not on confirming and comparing diel pattern of cyanobacterial species.

In the revised manuscript, we added more discussions in relation to prior literatures on cyanobacteria-microorganisms interactions Ln 423-427, 458-462, 488-490.

A minor point - there is no way the sequential filtering and washing of the Microcystis biomass could eliminate all the other microbiome bacteria, so in the Methods please describe the MCY and MIB as enrichments. The paper is written so that it reads like the MCY and MIB preparations are pure cyanobacteria and bacteria-only fractions, and this is certainly not the case.

Response: The descriptions of MCY and MIB were revised according to the reviewer’s suggestion. Ln89, 92.

Reviewer 2 Report

The proliferation of toxic cyanobacteria blooms such as Microcystis in many water bodies around the world is a matter of concern.  Here the authors investigated whether gene expression in Microcystis and its microbiome is affected by the circadian rhythms. This is very relevant since a recent study by Dittmann and co-authors (published in this Journal, not cited here) recently showed a large rise in extracellular microcystin after darkening.

There are a few serious problems that should be addressed before further reviewing including:

  1. To distinguish light/dark (diel) regime and impacts on the gene expression and metabolism from circadian clock rhythms you must include experiments were the light or the dark periods were extended checking whether the “wave behavior” of the rhythms persist (thought the amplitude usually declines). This was not examined/performed here and thus you measured diel transcript abundance rather than circadian clock effects.
  2. The design of the experiments is rather strange. If you wish to examine diel effects why not collecting the samples from the lake itself and examine transcript abundances? For the sake of the combining of microbiome with the Microcystis colonies? You collected the cells and place them on ice-Why? Because of the transfer distance? Did you examine the impact of the chilling on the gene expression even after the acclimation in the mesocosm? What relevance does your data have to the lake conditions?
  3. You must provide the reader with the raw data of the expression. Otherwise we can’t address the quality of the data and your analysis/interpratation.
  4. We read in the Introduction, “However, studies on circadian activities in freshwater cyanobacterial species are rare”. One recent study in Lake Erie has reported diel metabolic functions in Microcystis [5]”. However, studies by Taton and co-authors (Nature Comm. 2020) and a few others, investigated, in depth, the metabolic impact of the circadian clock.
  5. The interaction between Microcystis and its surrounding microbiome is rather complex. Nutrients and carbohydrates exchange (or impact on their availability) are just one aspect of this interaction, not necessarily the interesting one. Several studies showed that allelopathic interactions play an important role. One example (out of many) is the studies by Weiss and co-authors that identified the active components and investigated their impact.

And many more but I think the message is clear   

Author Response

Reviewer #2

The proliferation of toxic cyanobacteria blooms such as Microcystis in many water bodies around the world is a matter of concern.  Here the authors investigated whether gene expression in Microcystis and its microbiome is affected by the circadian rhythms. This is very relevant since a recent study by Dittmann and co-authors (published in this Journal, not cited here) recently showed a large rise in extracellular microcystin after darkening.

Response: This work was cited in the revised manuscript as reference #50 (Guljamow et al., 2021).

There are a few serious problems that should be addressed before further reviewing including:

  1. To distinguish light/dark (diel) regime and impacts on the gene expression and metabolism from circadian clock rhythms you must include experiments were the light or the dark periods were extended checking whether the “wave behavior” of the rhythms persist (thought the amplitude usually declines). This was not examined/performed here and thus you measured diel transcript abundance rather than circadian clock effects.

Response: We agree. We changed “circadian expression” to “diel expression” in the revised manuscript to avoid confusion.

  1. The design of the experiments is rather strange. If you wish to examine diel effects why not collecting the samples from the lake itself and examine transcript abundances? For the sake of the combining of microbiome with the Microcystis colonies? You collected the cells and place them on ice-Why? Because of the transfer distance? Did you examine the impact of the chilling on the gene expression even after the acclimation in the mesocosm? What relevance does your data have to the lake conditions?

Response: (1) The aim of this study was not just to examine diel effects of microbial communities. Rather, we aimed to elucidate potential interactions within the microbial community, i.e., between cyanobacteria and their co-occurring microorganisms. In other words, this study aimed to examine whether the diel activity of cyanobacteria could elicit a coordinated diel gene expression in their co-occurring microorganisms and how responses of the microbiome in turn impact cyanobacterial metabolism. The design of the experiment, specifically, separation of the cyanobacteria and other microorganism, allowed us to answer these questions unambiguously.

(2) Yes. After sample collection, we placed samples on ice before transported back to lab (as indicated in Ln 74-76).

(3) No, we did not examine the impacts of chilling on gene expression. We were aware that the transportation on ice would potentially induce cold shock responses. However, after the transportation, the water samples were incubated at room temperature under natural light for 2 hours to allow the water to warm up back to the room temperature. Then we further incubated the samples for 3 days to allow microbial communities to adjust to the new environmental conditions before start collecting samples for analysis. Therefore, the potential disturbance to the microbial communities during the transportation would have minimum effect to our results.

(4) Due to practical reasons, this microcosms experiments could not be performed on site and at in situ conditions. We used microorganisms directly collected from Lake Erie to set up microcosm incubation experiments. This allowed us to examine activities of Lake Erie microbial communities under controlled conditions, which provided unambiguous insights into the interactions between Microcystis and cooccurring microorganisms. However, as for all microcosm experiments, as much as we would like to study the “original community under the in situ conditions”, the nature of this experimental approach will inevitably introduce biases. Nevertheless, our microcosm incubation setup allowed us to examine the roles of different key species in biogeochemical processes. In addition, the deviation from the “in situ conditions” would have little effect on our overall conclusion. The controlled microcosms used in our study allowed detection of diel variations of gene expression in the microbiome and Microcystis when they were separated or together. Such data were not obtainable without manipulation of the in situ community. However, we agree that it is important to acknowledge the limitations of the method (502-520). In the revised manuscript, more text was added there to make this aspect clearer.

  1. You must provide the reader with the raw data of the expression. Otherwise, we can’t address the quality of the data and your analysis/interpretation.

Response: Analysis of RNA-seq data using normalized expression data is commonly used in many previous studies (Frischkorn et al., 2018; Kolody et al., 2019; Hark et al., 2019) as raw data are not comparable between samples/treatments due to sequencing depth variations.

Frischkorn, K.R., Haley, S.T., Dyhrman, S.T., 2018. Coordinated gene expression between Trichodesmium and its microbiome over day–night cycles in the North Pacific Subtropical Gyre. ISME J. 12(4), 997-1007.

Harke, M.J., Frischkorn, K.R., Haley, S.T., Aylward, F.O., Zehr, J.P., Dyhrman, S.T., 2019. Periodic and coordinated gene expression between a diazotroph and its diatom host. ISME J. 13(1), 118-131.

Kolody, B.C., McCrow, J.P., Allen, L.Z., Aylward, F.O., Fontanez, K.M., Moustafa, A., Moniruzzaman, M., Chavez, F.P., Scholin, C.A., Allen, E.E., Worden, A.Z., 2019. Diel transcriptional response of a California Current plankton microbiome to light, low iron, and enduring viral infection. ISME J. 13(11), 2817-2833.

Raw reads of sequences were available in the National Center for Biotechnology Information (NCBI) short read archive database under BioProject number SRR13450943-SRR13450964.

  1. We read in the Introduction, “However, studies on circadian activities in freshwater cyanobacterial species are rare”. One recent study in Lake Erie has reported diel metabolic functions in Microcystis [5]”. However, studies by Taton and co-authors (Nature Comm. 2020) and a few others, investigated, in depth, the metabolic impact of the circadian clock.

Response: We agree with the reviewer that there are many studies about cyanobacteria circadian activities. Most of them has been focused only on cyanobacterial species from the marine environments. We think our statement that “However, studies on circadian activities in freshwater cyanobacterial species are rare. One recent study in Lake Erie has reported diel metabolic functions in Microcystis” was correct. To make this point clearer, we have modified the relevant text. Ln 56-58.

  1. The interaction between Microcystis and its surrounding microbiome is rather complex. Nutrients and carbohydrates exchange (or impact on their availability) are just one aspect of this interaction, not necessarily the interesting one. Several studies showed that allelopathic interactions play an important role. One example (out of many) is the studies by Weiss and co-authors that identified the active components and investigated their impact.

Response: We agree with the reviewer that the interactions between Microcystis and associated microorganisms are very complex. We also agree that algicidal bacteria-Microcystis interactions represent an important aspect of such interactions (Weiss et al., 2019)

However, our study did not intent to be inclusive. Rather, the design of the present study was to examine two very specific aspects among many potential Microcystis-bacteria interactions. Those were (1) whether the diel activity of cyanobacteria could elicit a coordinated diel gene expression in their co-occurring microorganisms and (2) how responses of the microbiome in turn impact cyanobacterial metabolism. Therefore, we focused on examining genes that had diel expression patterns and known to play important roles in communications/interactions between cyanobacteria and their microbiome. The results generated from the designed experiments were able to answer our aimed questions. However, we agree that our study has limitations, relevant discussion was included in the manuscript (Ln 502-520).

Round 2

Reviewer 1 Report

The authors still ignored the Davenport paper. They need to discuss their results in light of the previous diel study, identiying areas of areement, disagreement and discuss what the two results may be different. The Discussion must have a full paragraph comparing and contrasting their results with the earlier paper.

Author Response

Reviewer #1

The authors still ignored the Davenport paper. They need to discuss their results in light of the previous diel study, identifying areas of agreement, disagreement and discuss what the two results may be different. The Discussion must have a full paragraph comparing and contrasting their results with the earlier paper.

Response: Requested discussion was added. Ln 510-518.

The text is also copied below:

“This can partially explain the different sets of diel expressed Microcystis transcripts found in our study and a previous study in Lake Erie that directly sequenced the metatranscripts of total microbial community [5]. Both studies found Microcystis transcripts of photosystem II and toxin biosynthesis were enriched in daytime, however, our study also found photosystem I and carbon fixation were enriched during the day, opposite from the other study [5]. In addition, our study found transcripts of glycogen degradation, vitamin biosynthesis, and detoxifying reactive oxygen species were nighttime enriched; they were not identified as nighttime expressed genes by the other study  [5].”